# Impact of Gut Microbiome on Gut Permeability in Liver and Gut Diseases

**DOI:** 10.3390/microorganisms13061188

**Published:** 2025-05-23

**Authors:** Punnag Saha, Phillipp Hartmann

**Affiliations:** 1Department of Pediatrics, University of California San Diego, La Jolla, CA 92093, USA; pusaha@health.ucsd.edu; 2Division of Gastroenterology, Hepatology & Nutrition, Rady Children’s Hospital, San Diego, CA 92123, USA

**Keywords:** CLDs, ALD, MASLD, PSC, IBD, gut microbiome

## Abstract

Hepatobiliary and gastrointestinal conditions, including chronic liver diseases and inflammatory bowel disease, are associated with significant morbidity and mortality globally. While the pathophysiology and symptoms vary from one disease to another, aberrations of the gut microbiome with deleterious microbial products affecting the intestinal barrier are common in patients suffering from these diseases. In this review, we summarize changes in the gut microbiome associated with various disease states and detail their role in gut barrier disruption and in modulating disease progression. Further, we discuss therapeutic interventions and precision medicine approaches targeting the microbiome, which have shown promise in alleviating these chronic illnesses in mouse models and patients.

## 1. Introduction

Chronic liver diseases (CLDs) are often characterized by the progressive deterioration of the liver that not only hampers normal physiological functions but can also progress to disruption of the hepatic architecture and scar tissue formation or fibrosis. CLDs are one of the major contributors to global morbidity, as 1.5 billion people were estimated to be suffering from a form of CLD in 2020 [1]. CLDs encompass multiple etiologies, such as alcohol-associated liver disease (ALD), metabolic dysfunction-associated steatotic liver disease (MASLD), which includes metabolic dysfunction-associated steatohepatitis (MASH) as a more severe form, and chronic viral hepatitis, i.e., hepatitis B virus- and hepatitis C virus-related hepatitis, as well as autoimmune disorders, e.g., primary sclerosing cholangitis (PSC), primary biliary cirrhosis (PBC), and autoimmune hepatitis. Among these, MASLD is estimated as the top contributor to CLDs (59%), followed by the hepatitis B virus (29%), the hepatitis C virus (9%), and ALD (2%). In contrast, other liver diseases, including PBC, PSC, autoimmune hepatitis, and others, contribute to only 1% of cases worldwide [2]. If untreated, CLDs often progress to end-stage life-threatening conditions such as cirrhosis, hepatocellular carcinoma, and liver failure, which are responsible for two million deaths per year globally and account for approximately 4% of all deaths worldwide [3]. In the United States alone, over 4.5 million adults have been diagnosed with CLDs, as per the National Center for Health Statistics, with a mortality rate of 16.4 per 100,000 people, making it the 10^th^ most common cause of death in the overall US population [4].

Similarly, many studies indicate that patients with intestinal conditions, including inflammatory bowel disease (IBD), can have liver disease as well, ranging from being relatively benign to more advanced severity [5]. Similar to CLDs, the global prevalence of IBD also increased between 1990–2019, as shown by a recent report [6]. According to this study, approximately 4.9 million cases of IBD have been reported worldwide during this period, while the United States is leading in IBD prevalence (245.3 cases per 100,000 people). IBD encompasses multiple chronic gastrointestinal conditions that are attributed to complex interactions between genetic, immune, and environmental factors and the gut microbiome [7]. The two main types of IBD are Crohn’s disease (CD) and ulcerative colitis (UC) [7]. Importantly, many studies have reported that PSC has a close association with IBD, as almost 70% of patients with PSC have IBD, particularly UC, which is often referred to as the PSC–IBD phenotype [8,9]. Although IBD and CLD are pathologically separate, both diseases share some similar features, including low-grade systemic and tissue (hepatic or intestinal) inflammation, increased gut permeability, and alterations of the gut microbiome [2].

In this review article, we elucidate the intricate relationship between the gut microbiome and these pathological conditions. In addition, we describe different novel microbiome-based therapeutic interventions to ameliorate these diseases.

## 2. Changes in the Intestinal Microbiome in Liver and Intestinal Diseases

The human gastrointestinal tract harbors a specific flora of microorganisms that consists of prokaryotes (bacteria), eukaryotes (fungi and protozoa), archaea, and viruses. Although often used interchangeably, the term “gut microbiota” refers to the commensal microorganisms that reside in the intestine, whereas the term “gut microbiome” indicates the total genomic content of these microorganisms [10]. Due to the large surface area of the intestine and favorable anaerobic growth conditions, these microorganisms can initiate their colonization in the gut from birth. However, various factors, including age, diet, environmental exposures, pollution, and even socioeconomic status, also play a crucial role in reshaping the overall composition of an individual’s gut microflora [11,12,13,14]. Under normal physiological conditions, the gut microbiome exerts several beneficial roles on the host, including providing energy to enterocytes and colonocytes by short-chain fatty acid (SCFA) synthesis, boosting the host’s immunity, regulating intestinal tight junctions, and protecting against invading pathogens or pathobionts [15]. However, the “friendly” gut microbial niche can change significantly to a “foe” under diseased conditions, which may further contribute to overall disease severity. This change is known as “dysbiosis” and refers to a decrease in beneficial microbes and an increase in potentially pathogenic microbes, with the associated deleterious effects on the host [16,17,18]. In this review article, we will focus on different CLDs and gut diseases to better understand the microbial composition changes in humans under these diseased conditions. It is important to note that alterations of the gut microbiome in human disease research are often correlative rather than causal. In contrast, the causality of the disease can be better evaluated with microbiome-focused interventions (e.g., probiotics, fecal microbiota transplantation, etc.) and is best determined in preclinical models given the controlled nature of the studies.

### 2.1. Alcohol-Associated Liver Disease

Multiple studies showed that patients with ALD have a significantly different gut microbiome than healthy controls. In a study conducted by Mutlu et al. [19], patients with ALD showed a decreased relative abundance of the phylum Bacteroidetes and a concomitant increase in Proteobacteria in their mucosa-associated colonic microbiome. Furthermore, the authors observed an increased level of endotoxemia in adult patients with ALD compared with healthy controls, which was correlated with gut dysbiosis. In another study by Dubinkina et al. [20], the gut microbiome of 99 adult patients with ALD (with and without liver cirrhosis) and 60 external controls was analyzed by shotgun metagenomic sequencing. Patients without cirrhosis had an increased fecal relative abundance of *Klebsiella pneumoniae*, *Lactobacillus salivarius*, and *Citrobacter koseri*, whereas patients with cirrhosis had increased *Bifidobacterium longum*, *Streptococcus thermophilus*, *Streptococcus mutans*, and *Lactobacillus salivarius* compared with controls. Moreover, they observed a significantly decreased relative abundance of butyrogenic species from the Clostridiales order, with a concurrent increase in opportunistic pathogens from the Enterobacteriaceae family. A similar trend in the gut microbiome profile in ALD was noted in a recently published study by Litwinowicz et al. [21], where the authors assessed raw reads of 16S rRNA sequencing from 511 samples [122 adult patients with alcohol use disorder (AUD), 75 with ALD, 54 with MASLD, and 260 healthy controls]. Butyrogenic families such as Ruminococcaceae, Lachnospiraceae, and Oscillospiraceae were found to be significantly depleted in patients with ALD compared with the AUD cohort or healthy controls; this was associated with an increase in the endotoxin-producing Proteobacteria family.

Apart from the gut bacteriome, various recent studies also delved into the role of the fungal microbiome, or mycobiome, and the virome in ALD in adults [22]. A study by Lang et al. [23] showed that ALD was associated with lower fungal diversity in 15 patients with AUD and 59 patients with alcohol-associated hepatitis compared with 11 healthy controls, as detected by fungal-specific internal transcribed spacer 2 (ITS2) amplicon sequencing of the feces. Patients with AUD or alcohol-associated hepatitis also had a significantly increased relative fecal abundance of *Candida* compared with controls. These findings were further confirmed by another study where the *Candida* species *Candida albicans* and *Candida zeylanoides*, along with the fungal genera *Debaryomyces*, *Pichia*, *Kluyveromyces*, and *Issatchenkia*, were increased in 66 patients with AUD compared with healthy controls. Interestingly, the authors also noted that following two weeks of alcohol abstinence, the abundance of these taxa decreased significantly in the abstinent patients compared with prior abstinence [24]. Alterations of the gut virome have also been reported in ALD in a study conducted by Hsu et al. [25], where the virome signature in 62 patients with AUD showed a lowered abundance of *Propionibacterium*, *Lactobacillus*, and *Leuconostoc* phages when compared with control subjects. Interestingly, two weeks of alcohol abstinence in patients with AUD resulted in an increased abundance of these phages compared with the pre-abstinence state. Furthermore, patients with AUD and progressive liver disease had an increased relative abundance of phages that target *Enterobacteria* and *Lactococcus* species in relation to nonprogressive patients.

### 2.2. Metabolic Dysfunction-Associated Steatotic Liver Disease

MASLD is defined as increased hepatic steatosis with the presence of one or more of the cardiometabolic risk factors: overweight/obesity, (pre-)diabetes, arterial hypertension, hypertriglyceridemia, and low high-density lipoprotein [26]. MASLD is associated with intestinal dysbiosis, including reduced diversity and richness of the gut microbiome in adults and children [27,28,29]. In a study by Loomba et al., whole-genome shotgun sequencing was carried out using stool samples from 72 adult patients with biopsy-confirmed MASLD and none to moderate fibrosis and 14 patients with MASLD and advanced fibrosis [30]. Patients with MASLD and advanced fibrosis had an increased relative fecal abundance of Proteobacteria and a lower relative fecal abundance of Firmicutes compared with the group with milder disease. At the species level, the SCFA producers *Ruminococcus obeum*, *Eubacterium rectale*, and *Faecalibacterium prausnitzii* (*F. prausnitzii*) were less abundant in patients with advanced fibrosis [30]. In another study, 104 participants (aged between 18 and 65) with a body mass index (BMI) of 25 or higher were divided into control subjects or subjects with MASLD based on magnetic resonance imaging [31]. The authors determined 16S rRNA-based gut microbiome profiling using the stool samples, fecal SCFAs, and targeted metabolomics. Their results showed that bacterial genera, including *Faecalibacterium*, *Subdoligranulum*, *Haemophilus*, and *Roseburia*, were significantly decreased in the MASLD group relative to controls. In addition, the abundances of several SCFA-producing families, including Ruminococcaceae, Lachnospiraceae, and Pasteureuaceae, were decreased in patients with MASLD, which correlated with the significantly decreased fecal levels of acetate and butyrate in these patients [31].

Apart from the bacterial microbiome, the mycobiome also changes significantly in MASLD. A recent study by Demir et al. determined the fungal pattern in the stool samples from 16 healthy adult controls and 78 patients with MASLD by ITS2 sequencing [32]. Their results showed that several fungal taxa, including *Candida albicans*, *Mucor* sp., *Pichia barkeri*, *Cyberlindnera jadinii*, *Penicillium* sp., and *Babjeviella inositovora*, were positively associated with steatohepatitis or significant fibrosis, whereas unknown Saccharomycetales and *Malassezia* sp. were associated with metabolic dysfunction-associated steatotic liver (MASL) or none to mild fibrosis [32]. Interestingly, a fungal signature was recently identified that could reliably differentiate MASLD from ALD with a high area under the curve (AUC) of 0.93; the fungal signature consisted of *Scopulariopsis*, *Kluyveromyces*, *Malassezia restricta* (*M. restricta*), and *Mucor*. The genera *Saccharomyces*, *Kluyveromyces*, and *Scopulariopsis*, and the species *C. albicans*, *M. restricta*, and *Scopulariopsis cordiae,* were significantly increased in patients with ALD, whereas the genera *Kazachstania* and *Mucor* were significantly increased in the MASLD cohort [33].

Changes in the gut virome have been identified by Lang et al. [34]. The authors extracted RNA and DNA virus-like particles from the stool samples from 9 adult control subjects and 73 patients with MASLD (29 patients with a MASL as defined by a NAFLD activity score of 0–4; 44 patients with MASH as defined by a NAFLD activity score of 5–8 and/or cirrhosis). They observed that the overall viral diversity was significantly lower in MASH and cirrhosis compared with MASL or healthy controls. In addition, *Lactococcus* phages were less abundant in MASH and/or cirrhosis, whereas *Streptococcus* phages TP-J34, *Escherichia* phages, and *Enterobacteria* phages were significantly more abundant in MASH and/or cirrhosis versus MASL. However, some of the phages were found to be very specific to individuals and were detected only in a small number of samples.

### 2.3. Cholestatic Liver Disease

Cholestatic liver disease is caused by the dysregulation of bile formation or bile flow, leading to jaundice, pruritus, and fatigue in patients [35]. Acute subtypes of cholestasis include common bile duct stenosis, cholangitis, and drug-induced liver injury, among others, whereas chronic cases of cholestasis include PSC, PBC, secondary sclerosing cholangitis, and biliary atresia. PSC is one of the major progressive forms of chronic cholestatic liver disease that can cause inflammation, fibrosis, and characteristic stricturing of intrahepatic or extrahepatic biliary ducts [36]. Multiple studies indicate that patients with PSC have an altered gut microbiome compared with healthy controls; in particular, *Veillonella* and *Enterococcus faecalis* are often associated with PSC [37,38,39,40]. Kummen et al. performed 16s rRNA gene sequencing and detected reduced fecal bacterial diversity and a significantly increased abundance of the genus *Veillonella* in 85 adult patients with PSC compared with 263 healthy individuals in Norway [37]. Bajer et al. reported similar results, with decreased fecal bacterial diversity and a significantly higher relative abundance of *Rothia*, *Enterococcus*, *Streptococcus*, and *Veillonella* in 32 adult patients with PSC in relation to 31 healthy controls in Czechia [38]. Of note, a pediatric study on PSC in Japan by Iwasawa et al. evidenced similar findings, with 27 children with PSC harboring a higher relative fecal abundance of *Enterococcus faecalis*, *Streptococcus parasanguinis*, and *Veillonella* species than 23 healthy controls [41]. In a recently published study by Özdirik et al., the authors assessed the gut microbiome obtained from the stool samples of 105 adult patients with PSC and 68 healthy controls from Germany and identified an increased relative abundance of *Enterococcus faecalis*, as well as its virulence factors, cytolysin and gelatinase, in patients with PSC [42].

Recent studies indicate that the gut mycobiome also plays a key role in patients with PSC [22,43,44,45]. A study by Lemoinne et al. was the first to determine the mycobiome signature in stool samples from 22 adult patients with PSC relative to 30 healthy subjects using ITS2 sequencing [43]. They observed that patients with PSC showed a significantly altered fungal diversity and composition compared with healthy subjects. This was supported by an increased proportion of *Exophiala* and a concomitant decrease in *Saccharomyces cerevisiae* proportions. In a separate study by Rühlemann et al., the authors conducted PCR and ITS2 sequencing to analyze the mycobiome signature in stool samples from 65 adult patients with PSC (including a subgroup of 32 patients with PSC–IBD) and 66 healthy controls [44]. However, the authors did not find a significant difference in fungal diversity in patients with PSC versus healthy controls as opposed to the earlier study by Lemoinne et al. [43]. However, the authors observed a significantly elevated abundance of the genera *Saccharomyces*, *Candida*, and *Dipodascus* and did not observe *Exophiala* in patients with PSC.

### 2.4. Inflammatory Bowel Disease

IBD is known to be a chronic inflammatory gastrointestinal condition that can develop in genetically susceptible hosts via a complex interplay between environmental, microbial, and immune-mediated factors [46]. UC and CD are the two major IBD subtypes. Multiple studies suggested that gut microbial alterations play a significant role in IBD etiology [47,48,49], supported by the fact that patients with IBD contain a significantly altered gut microbiome compared with healthy individuals [50,51,52]. Halfvarson et al. conducted a long-term study with 128 individuals (49 with CD, 60 with UC, four with lymphocytic colitis, 15 with collagenous colitis, and nine healthy controls), where samples were collected at 3-month intervals for 2 years [53]. Patients with IBD had decreased fecal microbial diversity and richness versus healthy controls, as well as beneficial gut commensals, including *Prevotella copri* and the butyrogenic bacterium *Faecalibacterium prauznitzii*, which were decreased in relative abundance in the IBD cohort.

In another study by Lloyd-Price et al., the authors analyzed the gut microbial profile obtained from 132 participants (three pediatric sub-cohorts and two adult cohorts) over 1 year (27 non-IBD, 67 CD, and 38 UC) [54]. Apart from the bacterial dysbiosis in patients with IBD in general, patients with CD had a decreased fecal relative abundance of obligate anaerobes such as the SCFA producers *F. prausnitzii* and *Roseburia hominis* and a concomitant increase in *Escherichia coli* that led to reduced butyrate levels compared with controls.

Apart from the bacteriome, the significant role of the mycobiome in patients with IBD has been established in multiple studies, where an increased abundance of various taxa from *Candida* has been associated with IBD severity [55,56,57,58,59]. Decreased fungal diversity is frequently observed in IBD [55]. Hoarau et al. found an increased relative fecal abundance of *Candida tropicalis* by ITS1 sequencing in 20 adult patients with CD compared with their 28 cohabiting relatives without CD [56]. In a separate study by Sokol et al., the authors reported an increased Basidiomycota/Ascomycota ratio and an increased proportion of *Candida albicans* in adult patients with 235 IBD relative to 38 healthy subjects [57].

Several studies also indicated the association between the gut virome and IBD pathogenesis, with a decrease in viral diversity [60,61,62]. Patients with IBD showed an increased abundance of *Clostridiales-*, *Alteromonadales-*, and *Clostridium acetobutylicum*-infecting phages and the *Retroviridae* family in comparison with healthy subjects [63]. Moreover, the enrichment of *Caudovirales* bacteriophages was confirmed in children with IBD [64] and adult patients with UC [65]. Cao et al. obtained terminal ileum biopsies and demonstrated decreased viral richness and increased *Gammatorquevirus*, a eukaryotic viral genus from the *Anelloviridae* family, in 103 subjects with CD versus 105 healthy individuals [66].

### 2.5. Common Findings

Assessing the microbiome signatures in patients (adults or children) with gut and liver diseases vs. healthy controls, it is very evident that these diseases cause dysbiosis, with an increased enrichment of pathobionts (e.g., *Klebsiella*, *Enterococcus*, *Veillonella*, etc.) and a decreased abundance of gut commensals [67]. These diseases decrease the overall bacterial diversity in the gut [68]. Certain beneficial bacteria, especially the SCFA producers (e.g., *F. prausnitzii*, *Roseburia*, *Ruminococcus*, etc.), are shown to be affected in ALD, MASLD, and IBD. Similarly, the mycobiome signature is also greatly reshaped in the underlying liver and gut diseases. Among the population of gut fungi, various species of *Candida*, especially *Candida albicans*, have been linked to ALD, MASLD, PSC, and IBD. Consequently, the composition of the core intestinal virus population (eukaryotic viruses and bacteriophages) also changes, which potentially plays a role in the progression of these diseases [69]. However, the exact conclusion of these virome changes is still unclear and needs further investigation. The common alterations of the bacterial, fungal, and viral microbiome in ALD, MASLD, PSC, and IBD have been summarized in Figure 1.

## 3. Role of Microbial Metabolites and Toxins in Liver and Gut Diseases

Intestinal permeability, one of the key determinants of the progression of liver and gut diseases, is affected by multiple factors (age, stress, diet, medication, alcohol intake, and the gut microbiome) [70]. Apart from the gut microbiome itself, several microbial metabolites and toxins also influence gut barrier function in disease states. In recent years, the role of various microbial metabolites, such as SCFAs, including acetate, butyrate, and propionate [71,72,73], endogenous ethanol [74], bacterial toxins (e.g., lipopolysaccharides) [75,76], and fungal toxins (e.g., β-D-Glucan and candidalysin) [77,78,79], has been extensively explored in the progression of CLD and intestinal diseases. While SCFAs play a protective role in gut health, other microbial products (e.g., lipopolysaccharides) can have detrimental effects. Under normal physiological conditions, a “healthy” gut microbiome synthesizes beneficial microbial metabolites and restricts the production of harmful toxins and metabolites. However, as described in the earlier section, an underlying disease is frequently associated with dysbiosis in the gut, resulting in the suppressed synthesis of beneficial metabolites and the increased production of microbial toxins. In this article, we will summarize the roles of these major microbial metabolites and toxins and their underlying mechanisms that contribute to gut barrier dysfunction, exacerbating disease (Figure 2).

### 3.1. Short-Chain Fatty Acids

SCFAs are one of the major microbial metabolites produced in the intestinal lumen by the gut commensals using soluble and insoluble dietary fibers [80]. Once produced, they are taken up from the lumen and transported to other organ systems via the bloodstream; however, they also modulate the gut barrier and serve as an energy source locally [81]. For instance, butyrate is an energy source for colonocytes [82] and is oxidized to produce ATP and CO_2_ [83]. Apart from that, SCFAs have a major role to play in boosting the intestinal immunity. They modulate the functioning of Th1 and Th17 cells to produce interleukin-22 (IL-22) and IL-17 cytokines, which can prevent the colonization of any invading pathogens [84]. In addition, butyrate can significantly decrease neutrophil recruitment, thereby lowering the production of proinflammatory cytokines, e.g., IL-6, tumor necrosis factor-α (TNF-α), and interferon γ (IFN-γ), and chemokines, including CC-chemokine ligand 3 (CCL3), CC-chemokine ligand 4 (CCL4), and chemokine (C-X-C motif) ligand 1 (CXCL1) [85].

Among the different SCFAs found in the gut, butyrate is the most abundant and is produced by several bacterial genera, including *Faecalibacterium*, *Roseburia*, *Ruminococcus*, *Eubacterium*, *Anaerostipes*, *Coprococcus*, *Subdoligranulum*, and *Anaerobutyricum* belonging to the phylum Firmicutes [86]. On the other hand, acetate is produced by different genera, e.g., *Bacteroidetes*, *Prevotella*, and *Bifidobacterium* [87], whereas propionate is produced by comparatively fewer microbes, e.g., *Selenomonas ruminantium*, *Megasphaera elsdenii*, *Bacteroides fragilis*, *Bacteroides vulgatus*, *Prevotella ruminicola*, and *Propionibacterium* [88].

ALD is associated with the reduced production of SCFAs in the lumen due to gut dysbiosis in both humans and preclinical animal models [89,90]. Cresci et al. used a murine model of acute ethanol-induced liver injury and butyrate supplementation to study the therapeutic potential of butyrate on intestinal tight junctions (TJs) [91]. Their results showed that ethanol exposure (chronic, short-term, or acute) can cause the decreased expression of TJ proteins, such as zonula occludens-1 (ZO-1) and occludin, as well as both the butyrate receptor GPR109A and transporter SLC5A8 in the colon. In another study by the same group, the authors used a chronic ethanol feeding model with a final ethanol binge [5% *v*/*v* ethanol-containing diet for 10 days, followed by single ethanol gavage (5 g/kg) 9 h before euthanasia] in mice to detect the effect of butyrate supplementation [92]. They observed that ethanol treatment in mice reduced colonic and intestinal TJ protein expression (claudin-3, occludin, and ZO-1). Ethanol treatment on Caco-2 monolayers decreased transepithelial electrical resistance. Butyrate treatment alleviated the effects of ethanol in both the in vivo and in vitro studies [92]. Similar to butyrate, the protective role of propionate in ALD has been established [93]: Propionate supplementation can prevent ethanol-induced hepatic steatosis and damage (marked by decreased ALT and AST levels). On the other hand, it can also boost the production of other SCFAs in the intestine, reduce intestinal inflammation, and increase the expression of TJ proteins (i.e., claudin-1, occludin, E-cadherin, and ZO-1), leading to improvements in gut barrier integrity in the ethanol-fed mice. Similar to ALD, the protective roles of SCFAs in MASLD have also been studied in both human [28] and animal models [94,95], indicating that decreased SCFA levels are one of the key contributors to disease development. Increased gut permeability and the decreased abundance of SCFA-producing gut commensals are also very common in patients with MASLD [31,96,97]. SCFAs can alleviate MASLD through various mechanisms [96,98], and SCFAs can increase gut barrier integrity by increasing hypoxia-inducible factor 1 (HIF-1) expression in the epithelial cells [99]. In addition, butyrate also upregulates the expression of several TJ proteins, such as claudin-1 and ZO-1, via the Akt-mediated pathway [100]. Reports showed that the binding of SCFAs to GPR43 receptors on colonic cells can stimulate K+ efflux, leading to hyperpolarization and inflammasome activation, which plays a protective role against colitis [101]. Moreover, SCFAs can be absorbed directly from the lumen and translocate to the liver via the portal vein [102]. They can decrease steatosis and inflammation through upregulation of the AMPK/Akt/Nrf2 signaling pathway [103]. Similarly, decreased SCFA production is also well-reported in PSC [104,105] and IBD [106,107].

### 3.2. Lipopolysaccharides

High serum levels of lipopolysaccharides (LPSs or endotoxins) have been observed in patients with ALD [108,109]. This is associated with increased gut permeability due to alcohol intake [110]. Many animal studies also reported the same in both acute and chronic ethanol-feeding models [111,112]. Increased gut permeability, combined with gut dysbiosis and portal endotoxemia, often potentiates liver injury via the toll-like receptor 4 (TLR4) signaling pathway [113]. Similar to ALD, LPS also contributes to MASLD progression, as shown in humans and preclinical studies [114]. Dysbiosis in patients with MASLD causes the increased enrichment of LPS-containing taxa in the gut [115]. In parallel, increased gut permeability contributes to the increased translocation of LPSs to the liver in MASLD [70]. LPSs can bind to LPS-binding proteins and can be recognized by the pattern recognition receptor TLR4, which is present on the cell membranes of hepatocytes and Kupffer cells. This leads to the activation of the proinflammatory TLR4–molecule myeloid differentiation factor 88 (MyD88)–nuclear factor-κB (NF-kB) pathway, resulting in the increased expression of proinflammatory cytokines, including TNF-α and IL-6 [116]. Increased levels of systemic endotoxemia have also been correlated with disease advancement in patients with IBD [117,118]. Systemic LPS levels can cause chronic low-grade inflammation in these patients, mediated by inflammatory cytokines (e.g., TNF-α and IL-6), as mentioned above. Similarly, increased intestinal inflammation can trigger the TNF-α-mediated NF-κB pathway. This results in decreased ZO-1 protein expression and increased myosin light chain kinase (MLCK) expression, which increases intestinal permeability and helps LPSs to translocate from the gut to the circulation [119].

### 3.3. Fungal Toxins

Apart from bacterial LPSs, recent studies have also highlighted the significant role of fungal toxins, including candidalysin and β-glucan, in liver disease conditions, particularly in ALD [79,120]. Yang et al. [120] showed that chronic alcohol administration in mice resulted in mycobiota enrichment, which was also detected in patients with ALD. This resulted in the increased translocation of β-Glucan into the bloodstream, which further contributed to hepatic inflammation and liver injury via C-type lectins, including the receptor CLEC7A (also known as Dectin1), on Kupffer cells. Separately, Chu et al. reported that the fecal mycobiome from patients with ALD had an enriched proportion of candidalysin-producing *C. albicans* [79]. Furthermore, they also showed that oral administration of candidalysin-positive *C. albicans* exacerbated the ethanol-induced liver injury. However, the deleterious effects of candidalysin were independent of the CLEC7A receptor in mice, suggesting the possible involvement of other mechanisms. In addition, candidalysin alone did not affect intestinal permeability, indicating that another hit (e.g., ethanol or a toxic microbial product) is required for the translocation of fungal toxins into the circulation. Further, gavage of other fungal populations, including *M. restricta*, can exacerbate experimental ethanol-induced liver injuries in mice via Kupffer cells through C-type lectin domain family 4, member N (Clec4n) signaling [121].

So far, no major studies have been conducted to ascertain the direct role of mycotoxins, including candidalysin or β-glucan, in mouse models of MASLD or other chronic liver diseases. Candidalysin promotes the activation of the NLRP3 inflammasome in phagocytes [122] and neutrophil recruitment, and concomitantly increases proinflammatory cytokine (including IL-1α, IL-1β, and IL-8) release [123]. Therefore, currently, it can only be speculated that these mycotoxins could play a significant role in the progression of other liver and gut diseases, and further studies are needed to explore this.

### 3.4. Endogenous Ethanol

Alterations of the gut microbiome are also associated with increased endogenous ethanol production in patients with MASLD [124] and animal models [125]. In a study by Yuan et al. [126], the authors observed an increased abundance of the ethanol-producing *Klebsiella pneumoniae* strain in 60% of patients with MASLD. Furthermore, the transplantation of the bacteria in mice showed distinct MASLD histopathology, with increased levels of hepatic triglyceride (TG) and serum biomarkers such as alanine transaminase (ALT) and aspartate transaminase (AST). Mechanistically, ethanol, along with its metabolite acetaldehyde, can directly affect the gut barrier. As shown in an in vitro study, a combination of both ethanol and acetaldehyde increased paracellular permeability due to decreased levels of ZO-1 and occludin at the intercellular junctions of Caco-2 spheroids without causing significant cytotoxicity [127]. Furthermore, a study by Zong et al. demonstrated that increased levels of endogenous ethanol in mice fed a high-fat diet led to the increased catalytic activity of cytochrome P450 family 2 subfamily E polypeptide 1 (CYP2E1) in the liver, which is profibrotic [128]. This increase in the enzymatic activity of CYP2E1 to oxidize ethanol causes the formation of acetaldehyde and depletes the cellular antioxidant glutathione (reduced form), leading to the generation of highly reactive free radicals in the liver [74,129].

### 3.5. Bile Acids

The liver synthesizes primary bile acids through both classical [cholesterol 7α-hydroxylase (CYP7A1)-dependent] and alternative pathways [cholesterol 7α-hydroxylase (CYP7A1)-dependent], using cholesterol as a substrate. These primary bile acids, e.g., cholic acid and chenodeoxycholic acid (in humans), are then transported by the bile acid ducts to the intestine. Approximately 95% of the primary bile acids are then reabsorbed in the intestine, especially in the ileum, and transported back to the liver through the enterohepatic circulation. However, the remaining 5% of primary bile acids then enter the colon, where the gut bacteria convert these to secondary bile acids, e.g., deoxycholic acid and lithocholic acid [130,131]. Therefore, the gut microbiome acts as a key contributor to the host’s bile acid metabolism and regulation.

Previously, clinical studies showed that patients with MASH and fibrosis have significantly increased serum/plasma total bile acid levels [132,133]. Similarly, patients with PSC had increased levels of bile acids, conjugated fractions, and primary-to-secondary bile acid ratios relative to healthy controls [134]. Moreover, serum bile acid levels with clinical biomarkers have also been proposed as a potential predictor in patients with liver fibrosis [135,136]. Mechanistically, secondary bile acids activate the Farnesoid X receptor (FXR) [order of affinity: chenodeoxycholic acid > lithocholic acid = deoxycholic acid > cholic acid] and the Takeda G protein-coupled receptor 5 (TGR5) present in the intestine. Activation of FXR negatively regulates hepatic bile acid secretion and plays a significant role in glucose and lipid metabolism [137,138]. An increase in deoxycholic acid, and concomitant decreases in FXR-agonistic chenodeoxycholic acid, are also reported in patients with MASLD, indicating the importance of the bile acid–FXR axis [139]. Of note, bile acid binding can improve diet-induced steatohepatitis in mice [140]. Increased serum levels of bile acid and fibroblast growth factor 19 (FGF19), which regulates bile acid synthesis, were detected in patients with alcohol-associated hepatitis [141]. Another study also reported that patients with AUD had increased levels of deoxycholic acid in their serum [142]. FXR activation has been shown to abrogate NLRP3 inflammasome-mediated caspase 1 activation in macrophages [143]. On the other hand, FXR activation also inhibited the TGF-β/SMAD pathway in hepatic stellate cells, causing decreased hepatic fibrosis in mice [144]. In addition, secondary bile acid derivatives also participate in increasing the differentiation of intestinal Treg cells, which modulates the overall balance of Th17 and Treg in the intestine [145]. Another study also showed that consumption of a high-fat diet caused an increased production of deoxycholic acid in mice, which promoted macrophage polarization to the M1 phenotype, leading to colonic inflammation [146].

### 3.6. Indoles

Indoles and indole-related compounds (e.g., indole-3-pyruvate, indole-3-lactate, indole-3-propionate, indole-3-acetate, tryptamine, etc.) are generated from the essential amino acid tryptophan by gut commensals using the enzyme tryptophanase. Interestingly, these compounds are only produced in the small intestine by the gut microbiome as mammals lack the enzyme [147]. Several bacterial genera, including *Escherichia coli*, *Clostridium*, *Bacteroides*, and *Proteus vulgaris*, are known indole producers in the gut [148]. There are several beneficial effects exerted by indoles and indole-related compounds in the intestinal microenvironment. Mechanistically, these compounds bind to the aryl hydrocarbon receptor (AhR), which is present on the intestinal epithelial cells and immune cells [149]. This binding to AhR causes the expression of downstream anti-inflammatory cytokines, e.g., IL-22 and IL-17, which help to maintain gut homeostasis [150]. Another study also showed that the colonization of indole-producing bacteria in geriatric mice enhanced goblet cell differentiation and IL-10 expression [151]. In addition, the indole compound tryptamine also increases gastrointestinal motility by inducing serotonin release [152]. In vitro, treatment with a physiologically relevant amount of indoles in the human enterocyte cell line HCT-8 showed an increase in transepithelial resistance and a decrease in NF-kB-mediated inflammation, as well as the attachment of pathogenic *Escherichia coli* to the cells [153]. Clinical studies reported that patients with MASLD have a significantly decreased amount of indoles in their circulation compared with lean subjects [154]. On the other hand, analysis of stool samples from patients with alcohol-associated hepatitis also showed decreased levels of indole-3-acetate and indole-3-lactate [155].

## 4. Therapeutic Interventions to Alleviate Liver and Gut Diseases

Several gut microbiome-targeted therapeutic approaches have been conceptualized and tested over the past few years to improve hepatobiliary and gastrointestinal diseases. This section summarizes (Table 1) and briefly discusses these major microbiome-targeted therapeutic interventions. This section has been further categorized based on untargeted and targeted microbiome-based therapeutic interventions.

### 4.1. Untargeted Microbiome-Based Therapeutic Approaches

#### 4.1.1. Fecal Microbiota Transplantation

Fecal microbiota transplantation (FMT) is an approach that delivers gut microbes from healthy donors to individuals with disease. This transplantation aims to reconstruct the recipient’s overall gut microbiome in such a way that it becomes more diverse, with beneficial gut commensals, which decreases the abundance of pathobionts, leading to improved functionality and benefiting health. The efficacy of FMT has been determined in both humans and preclinical models in diverse disease conditions, ranging from metabolic conditions [156] and autoimmune diseases [157] to neurological problems [158]. Craven et al. performed FMT via a randomized controlled trial (RCT) in 21 adult patients with MASLD and followed up at 2 weeks, 6 weeks, and 6 months post-FMT [159]. They observed that FMT could help reduce gut permeability in patients with MASLD. However, it did not decrease insulin resistance in these patients. In another RCT study by Xue et al., 75 adult patients with MASLD were divided into non-FMT (28 patients) and FMT (47 patients) groups and received probiotics and FMT treatment, respectively [160]. After 1 month, the authors observed that FMT improved overall gut dysbiosis in the patients with MASLD, with higher bacterial diversity and the enrichment of bacterial genera, including *Ruminococcus* and *Prevotella*. However, they found no significant difference in blood lipid and liver function results between the FMT and non-FMT groups. In addition, the authors observed that the beneficial effects of FMT were more pronounced in lean MASLD patients compared to subjects with MASLD and obesity. In a recently published study by Bajaj et al., the authors conducted a phase 2, double-blind, randomized, placebo-controlled trial in 60 adult patients with cirrhosis and hepatic encephalopathy with a 6-month follow-up [161]. They observed that FMT, regardless of route of delivery (oral or enema) and number of doses (one through three), was safe for these patients, and the recurrence of hepatic encephalopathy was decreased in patients who received FMT compared with those who received a placebo treatment. FMT treatment also increased the abundance of butyrogenic bacteria and decreased pathobionts in the gut. Allegretti et al. conducted an open-label pilot study with a single FMT treatment via colonoscopy in 10 adult patients with PSC–IBD (nine patients had UC; one patient had CD) and followed up until 6 months [162]. The authors noted that three out of the ten patients who received FMT had a significant reduction (≥50%) in blood alkaline phosphatase (ALP) levels that correlated with a significant improvement in gut microbial diversity. Over the past few years, many RCTs have been conducted to assess the efficacy of FMT in IBD patients [163,164,165,166,167]. The overall results showed promising aspects, with significantly better endoscopic and histological remission that correlated with the enrichment of a beneficial gut microbiome in these patients.

#### 4.1.2. Prebiotics, Probiotics, Synbiotics, and Postbiotics

By definition, probiotics are ingestible and viable forms of beneficial bacteria, while prebiotics are indigestible polysaccharides that act as a nutritional source for these beneficial bacteria, thereby enhancing their growth in the gut. Synbiotics are a combination of both probiotics and prebiotics, whereas postbiotics are bioactive microbial compounds that help the host’s health [168]. In particular, the beneficial effects of prebiotics and probiotics have been explored by numerous preclinical and clinical studies over the years under various disease conditions [168,169]. Prebiotics (e.g., inulin, cellulose, pectin, and lignin) cannot be broken down by the host. Therefore, gut commensals can ferment these polysaccharides to produce SCFAs, which in turn improve intestinal immunity [170]. In addition to SCFA production, prebiotics can participate by influencing the host’s immune health through other mechanisms, as shown by a preclinical study using only prebiotic treatments in obese and diabetic mice. The results showed that prebiotic treatments in these mice induced the endogenous production of glucagon-like peptide-2 (GLP-2) with improved intestinal tight junction integrity and helped to decrease plasma LPSs and systemic and hepatic inflammatory cytokine levels [171].

Similarly, the beneficial effects of probiotics have also been deciphered. Preclinical studies showed that *Lactobacillus plantarum* can induce epidermal growth factor receptor-mediated protection of colonic epithelial TJ proteins and decrease pathological lesions, endotoxemia, TG deposition, and oxidative stress in the livers of a murine model of ALD [172]. In another study, the use of probiotics (*Bifidobacterium*, *Lactobacillus bulgaricus*, and *Streptococcus thermophilus*) showed decreased levels of liver disease markers, plasma ALT, AST, hepatic triglycerides, and reduced alcohol-mediated apoptosis by suppressing FOXO1 in a murine model of ALD [173]. A recent meta-analysis analyzed nine RCTs and found that probiotic treatment improves ALD by significantly decreasing serum ALT, AST, and GGT, but it does not significantly improve total bilirubin or inflammatory cytokine levels (TNF-α and IL-6) [174]. In addition, treatment with synbiotics also showed amelioration of ethanol-induced colonic oxidative stress, inflammation, and increased expression of TJ proteins both in vivo and in vitro in rats [175]. Several studies have also been conducted to detect the efficacy of probiotics in animal models of diet-induced steatohepatitis [176,177] and patients with MASLD [178,179]. A recently published meta-analysis included 34 RCTs to assess the efficacy of probiotics, prebiotics, and synbiotics in patients with MASLD [180]. The authors detected that these therapeutic approaches exerted improved clinical results in these patients, with decreased circulatory levels of liver enzymes, improved lipid profiles, and decreased inflammatory cytokine levels. However, they also found that these intervention strategies did not decrease hepatic steatosis (measured by liver ultrasound) or LPS levels significantly. Vleggaar et al. conducted an RCT in 14 adult patients with PSC–IBD using probiotics comprising six strains (*Lactobacillus acidophilus*, *Lactobacillus casei*, *Lactobacillus salivarius*, *Lactococcus lactis*, *Bifidobacterium bifidum*, and *Bifidobacterium lactis*) for 3 months [181]. However, their data did not show any significant improvement in these patients, as bilirubin and liver function test markers (ALP, GGT, ALT, and AST) did not decrease significantly. In contrast, several RCTs have been conducted involving patients with IBD, employing probiotics with a single bacterial strain [182,183] or multiple strains [184,185], prebiotics [186,187], and synbiotics [188,189], which showed overall improved clinical outcomes in these patients, with a better quality of life. In addition, postbiotics such as SCFAs, vitamins, proteins, bacteriocins, and amino acids have several beneficial roles in modulating the host’s health by different mechanisms under normal physiological conditions and have shown promising results in preclinical models in various liver and gut diseases [190,191,192]. However, RCTs are needed to further establish their therapeutic potential in humans against these diseases.

#### 4.1.3. Fungi-Focused Interventions

As described in the earlier sections, CLDs also lead to dysbiosis of the mycobiome and an increased translocation of fungal toxins and metabolites into the circulation. Therefore, antifungal treatment interventions have been explored in ethanol-induced steatohepatitis by Yang et al. [120]. Oral administration of the antifungal amphotericin B helped to decrease fungal overgrowth and β-glucan translocation in ethanol-fed mice. Moreover, amphotericin B treatment also protected the mice from ethanol-induced steatohepatitis [120]. A separate study utilized a humanized mouse model of MASLD, in which germ-free mice were transplanted with cells from patients with steatohepatitis and fed a Western diet [32]. Treatment with amphotericin B in these mice resulted in lower ALT, hepatic TG, and cholesterol levels compared with those only fed a Western diet. Additionally, the gene expression of inflammatory cytokines and fibrotic markers was reduced in the amphotericin B-treated mice, indicating the therapeutic potential of the antifungal in underlying liver disease conditions. An RCT aimed to assess the efficacy of oral fluconazole therapy (200 mg daily for 3 weeks) in 68 *Candida*-positive adult patients with UC [193]. After exclusions, 61 patients were divided into fluconazole (31 patients) and placebo (30 patients) treatment groups. After 4 weeks, fluconazole-treated patients with UC showed significantly decreased fecal calprotectin levels and improved histological scores compared with the placebo, indicating promise for clinical use in IBD. In addition, fluconazole was well-tolerated in these patients, and no drug-induced toxicity was noted.

### 4.2. Targeted Microbiome-Based Therapeutic Approaches

#### 4.2.1. Farnesoid X Receptor (FXR) Agonists

FXR is a transcriptional regulator activated by bile acids or FXR agonists and is highly abundant in the liver and the intestines [137]. Enterohepatic recirculation of bile acids activates FXR, which then dimerizes with the retinoid X receptor (RXR) inside the nucleus and starts the gene expression of FXR response elements [194]. This leads to increased expression of FGF19 (in humans) and FGF15 hormones (in mice), which reach the liver via the portal circulation and regulate bile acid synthesis, glucose, and lipid metabolism in the liver [138,195,196]. Therefore, FXR activation acts as one of the master negative feedback regulators of hepatic bile acid synthesis and lipid biosynthesis when a high amount of bile acids is present. Mechanistically, FXR inhibits NF-κB activation and thereby decreases the production of inflammatory cytokines and chemokines, as well as fibrosis [197,198].

Multiple RCTs have been conducted with various FXR agonists due to their significant therapeutic potential [199,200,201]. Obeticholic acid has been trialed in several CLDs, including PBC [202,203,204], MASLD [205,206], and PSC [207]. In patients with PBC who do not respond to ursodeoxycholic acid treatment, obeticholic acid can be considered as a second-line therapy as it improves ALP and total bilirubin levels compared with the placebo group [202,208]. In adult patients with MASLD, obeticholic acid treatment showed significant improvement in fibrosis and liver histology in two large phase 2 and phase 3 RCTs [205,206]. Separately, obeticholic acid treatment (up to 5–10 mg) in adult patients with PSC showed a significant dose-dependent decrease in ALP levels (14–25%) in a phase 2 clinical trial [207]. However, total bilirubin levels in these patients did not decrease compared with the placebo-treated group. Another promising FXR agonist, Cilofexor, has also been trialed in patients with MASH and advanced fibrosis [209], as well as in patients with PSC [210]. Combination therapy with Cilofexor (30 mg) and Firsocostat (18 mg) for 48 weeks showed a non-significant NAFLD activity score and fibrosis in adult patients with MASH [209]. In adult patients with PSC, 12-week treatment with Cilofexor (100 mg) resulted in significant reductions in serum biomarkers, including ALP, γ-glutamyl transpeptidase (GGT), ALT, and AST [210]. So far, only preclinical studies have been conducted to elucidate the mechanisms of FXR agonists in ALD [211,212,213,214], which showed that interventions with these agonists can protect against ethanol-induced steatohepatitis in mice [215,216]. Similarly, FXR activation by the agonist INT-747 protected the intestinal barrier, increased goblet cell function, and prevented colon shortening in the dextran sodium sulfate-treated colitis model [217]. However, RCTs are required to further evaluate these findings in patients.

#### 4.2.2. Bioengineered Bacteria

The application of bioengineered bacteria represents a novel gut bacteriome-targeted therapeutic approach currently under exploration. A study by Hendrikx et al. demonstrated that chronic ethanol feeding in mice results in decreased production of the microbial metabolite indole-3-acetic acid (IAA), a finding confirmed in patients with alcohol-associated hepatitis [155]. IAA acts as one of the ligands for the aryl hydrocarbon receptor and regulates IL-22 production in the intestine. Moreover, reduced IL-22 expression leads to decreased production of the intestinal antimicrobial peptide REG3G and increased bacterial translocation to the liver in ethanol-fed mice. The application of an IL-22-producing bioengineered *Lactobacillus reuteri* strain showed protective effects in ethanol-fed mice, with increased *Reg3g* expression. Similar outcomes were observed with IAA supplementation. In another study by Kouno et al., the bioengineered *Escherichia coli* Nissle 1917 strain (EcN-Ahr) was used to produce tryptophan, which could be further converted to IAA [218]. Gavage of the EcN-Ahr strain to ethanol-fed mice led to reduced levels of serum ALT, hepatic TG, and inflammatory cytokine and chemokine gene expressions. Additionally, supplementation of the EcN-Ahr strain in these mice also enhanced *Reg3b* and *Reg3g* gene expression and increased IL-22-expressing type 3 innate lymphoid cells (ILC3) in the intestine.

#### 4.2.3. Precision Editing Against Specific Gut Pathobionts

Recently, another promising microbiome-based therapeutic approach has been established where tungstate-mediated microbiome editing was used to attenuate colitis in mice [219]. The authors specifically targeted the Enterobacteriaceae family (phylum Proteobacteria), which increases significantly in colitis-induced dysbiosis. The use of tungstate treatment selectively targeted microbial respiratory pathways that are dependent on the molybdenum cofactor and are functional during gut inflammation. Furthermore, tungstate treatment did not affect the overall gut microbiome composition, proving its precision against the pathobiont, and also ameliorated colitis-mediated gut inflammation in mice.

#### 4.2.4. Phage Therapy

Phage therapy represents a novel precision medicine approach currently being employed to treat gastrointestinal diseases. Patients with alcohol-associated hepatitis exhibit an increased abundance of cytolysin-positive *Enterococcus faecalis* in the gut, which contributes to hepatocyte death and liver injury, correlating with heightened mortality compared with healthy controls or those with AUD [220]. Treatment with bacteriophages that specifically target cytolysin-positive *Enterococcus faecalis* strains significantly improved ethanol-induced steatohepatitis in gnotobiotic mice transplanted with stool from patients with alcohol-associated hepatitis. In another study, researchers specifically targeted the *Klebsiella pneumoniae* strain, which produces large amounts of ethanol and is enriched in patients with MASH [221]. Their results indicate that phage therapy reduces steatohepatitis-related hepatic dysfunction and inflammation and regulates lipid metabolism without impacting the functions of vital organs, such as the liver or kidneys. In fecal samples of patients with PSC, an increased abundance of *Klebsiella pneumoniae* and *Enterococcus gallinarum* was detected, which is correlated with increased disease severity and poor clinical outcomes [222]. Administration of a phage cocktail in gnotobiotic mice (transplanted with *Klebsiella pneumoniae* from a patient with PSC) and specific pathogen-free mice via both oral and intravenous routes significantly decreased the fecal abundance of *Klebsiella pneumoniae*, also showing decreased serum ALP levels, attenuation of liver inflammation, and fibrosis progression. These studies suggest that the selective elimination of pathobionts through phage therapy could be beneficial in treating patients with CLDs. However, randomized controlled trials (RCTs) with large cohorts are required to validate these findings in humans.
microorganisms-13-01188-t001_Table 1Table 1Therapeutic interventions targeting the intestinal microbiome in chronic gut and liver disease (selected articles). ↑ denotes increase, whereas ↓ denotes decrease.Type of InterventionDisease TypeIntervention DetailsOutcomeReferenceFecal microbiota transplantionMASLDRCT in 21 adult patients with MASLD; follow-up at 2 weeks, 6 weeks, and 6 months post-FMT.Reduced gut permeability.No effect on insulin sensitivity.Craven et al.[159]MASLD RCT in 75 adult patients with MASLD, who were divided into non-FMT (28 patients) and FMT (47 patients) groups; follow-up after 1 month.FMT improved overall gut dysbiosis in the patients with MASLD, with higher bacterial diversity.*Ruminococcus* and *Prevotella* ↑.No significant difference in blood lipid and liver function results.Xue et al.[160]ProbioticsALDMeta-analysis of 9 randomized controlled trials from 2008 to 2023(n = 639)Decreased levels of ALT, AST, and GGT.No effect on total bilirubin and inflammatory cytokines.*Bifidobacteria*, *Lactobacillus* ↑*Escherichia coli* ↓Xiong et al.[174]Probiotics, Prebiotics, and SynbioticsMASLDMeta-analysis of 34 randomized controlled trials until March 2024 (n = 12,682)Decreased levels of ALT, AST, ALP, GGT, and inflammatory cytokines.Decreased fibrosis.No effect on endotoxemia.Pan et al.[180]ProbioticsPSC–IBD14 adult patients with PSC–IBD treated with probiotics comprising six strains for 3 monthsNo significant improvement in liver function test.No changes in pruritus, fatigue, and stool frequency.Vleggaar et al.[181]FXR agonistsMASLDRandomized clinical trial in adult patients with non-cirrhotic MASH; obeticholic acid (25 mg daily) or the placebo was given orally for 72 weeks.45% of patients in the obeticholic acid group had improved liver histology (2-point or greater improvement in NAFLD activity score without worsening of fibrosis) compared with 21% of patients in the placebo group.Neuschwander-Tetri et al.[205]PSCRandomized clinical trial in 76 adult patients with PSC with a placebo or 2 doses of obeticholic acid once daily for 24 weeks; followed by a 2-year, long-term safety extension.Decreased serum ALP levels in patients treated with 5–10 mg of obeticholic acid.Kowdley et al.[207]ALDEvaluation of the FXR agonist fexaramine in mice with chronic alcohol-induced liver disease.Improved bile acid–FXR–FGF15 signaling and lipid metabolism.Decreased ethanol-induced liver disease in mice.Hartmann et al.[214]Bioengineered bacteriaALDBioengineered *Lactobacillus reuteri* strain in mice with chronic alcohol-induced liver disease.Increased production of IL-22 and IL-22-mediated signaling.Protection from liver injury induced by alcohol.Hendrikx, T., et al.[155]ALDBioengineered *Escherichia coli* Nissle 1917 strain (EcN-Ahr) in mice with chronic alcohol-induced liver disease.Decreased levels of serum ALT, hepatic TG, and inflammatory cytokine and chemokine gene expression.Increased *Reg3b* and *Reg3g* gene expression in the intestine.Kouno et al. [218]Precision editing of the gut microbiomeIBDInhibition of microbial respiratory pathways by tungstate in a murine model of colitis.Decreased gut inflammation, and protection from colitis.Zhu et al. [219]Fungi-focused interventionsALDOral administration of the antifungal amphotericin B in alcohol-induced liver disease.Decreased fungal overgrowth andβ-glucan translocation in the circulation.Yang et al. [120]MASHOral administration of amphotericin B in Western-diet-fed germ-free mice (transplanted with cells from patients with steatohepatitis).Decreased levels of serum ALT, hepatic TG, and cholesterol.Reduced hepatic inflammatory cytokine and chemokine gene expression.Demir et al. [32]IBDOral fluconazole therapy (200 mg daily for 3 weeks) in 68 *Candida*-positive adult patients with UC; follow-up after 4 weeks.Decreased fecal calprotectin levels in the patients.Jena et al. [193]Phage therapyALDBacteriophage treatment to target cytolysin-positive *Enterococcus faecalis* in murine model of alcohol-induced steatohepatitis.Alleviation from liver injury and steatohepatitis in mice.Duan et al. [220]MASLDPhage therapy against alcohol-producing *Klebsiella pneumoniae* in MASLD.Decreased hepatic inflammation and regulation of lipid metabolism in mice.No deleterious effects on the other vital organs.Gan et al. [221]PSCPhage cocktail treatment in gnotobiotic mice (transplanted with *Klebsiella pneumoniae*).Attenuation of liver inflammation and hepatobiliary injury in mice.Ichikawa et al. [222]


## 5. Existing Challenges and Future Directions

Despite tremendous scientific advances in the area of gut microbiome research, many limitations still exist. One of the major concerns remains whether gut dysbiosis is the cause of the gastrointestinal and liver diseases, the effect of these diseases, or whether there are any bidirectional dynamics in play. Gnotobiotic mouse models have helped to elucidate this point. Germ-free mice transplanted with stool from patients with ALD or MASLD and treated with a specific ethanol- or diet-induced steatohepatitis model, respectively, usually develop more liver diseases than conventional wildtype mice treated with the corresponding model, indicating that the human microbiome from patients with liver disease contributes to liver disease pathogenesis [223]. Further, gavaging selected microbes to mice with a liver disease-specific mouse model will indicate whether those microbes tend to worsen, improve, or not change the disease in a certain liver disease [224,225,226]. However, the results of therapeutic interventions, particularly if beneficial, should be confirmed in human trials.

There are also a few concerns regarding microbiome-targeted therapeutics, especially FMT [227]. In addition to clinical parameters to confirm the donor’s health, proper screening of the donor stool sample is required to eliminate any risk of pathobiont transmission, as well as the transmission of any potential virulent factors, e.g., antimicrobial resistance genes [228]. Furthermore, the ideal route for the delivery of FMT (upper gastrointestinal tract or lower gastrointestinal tract) to patients is still unclear. In addition, the type of samples (fresh or frozen) and sample preparation techniques vary from laboratory to laboratory [229]. Therefore, standardization is required to ensure maximum efficacy and the reproducibility of transplantation in patients. Assessing the long-term risks and safety parameters, and detecting the individual’s vulnerability to receiving these treatments, should be considered first. In addition, the identification of key strains, delivery strategies to ensure the colonization of the strains, and proper clinical evaluations are needed to assess the full potential of these treatments [230].

Similar challenges exist for prebiotic, probiotic, synbiotic, and postbiotic treatments in patients. The specific standardization of probiotic strains, doses, routes of delivery, quality control, and storage may vary greatly [231]. Therefore, optimizing these factors poses a huge challenge. In contrast, some of these factors are not specifically required for prebiotic treatments. However, for a prebiotic to function to its full potential, assistance from probiotic strains may also be needed [232]. Therefore, it may be more challenging to treat patients with prebiotics due to the decreased enrichment of beneficial microbes in the gut with the increased abundance of pathobionts. In addition, the clinical safety of these interventions in critically ill patients and immunocompromised individuals is a huge concern [233]. Another important issue is to find the correct duration for interventions in patients, as short-term treatments may fail to produce any significant impact [234].

While targeted microbiome-based therapeutic approaches have shown promising results, there are also many concerns. FXR agonists, e.g., obeticholic acid, have been reported to cause pruritus (77%; 149/193) and fatigue (33%; 63/193) in patients with primary biliary cholangitis. Additional gastrointestinal symptoms, including bloating, diarrhea, and abdominal discomfort, were also noted in these patients [204]. However, application of non-steroidal FXR agonists, e.g., Cilofexor, showed better clinical outcomes in patients with hepatobiliary diseases, with significantly decreased adverse effects, including pruritus (20–29% of Cilofexor-treated patients versus 15% of placebo-treated patients) [209]. Similarly, clinical risks associated with phage therapy, which include the potential onset of strong immunogenicity in the host, have not been fully explored [235]. In addition, other issues, for instance, targeting only a narrow range of bacteria, the development of phage-resistant mutant bacteria, and the production of anti-phage antibodies in the host, may serve as potential limitations for phage therapy [236]. Other targeted approaches, including precision editing of gut pathobionts and administration of bioengineered bacteria, have only been tested in preclinical models so far. Therefore, clinical trials with larger cohorts are needed to assess the therapeutic potential of these methods in patients. Moreover, the clinical efficacy, safety, and standardization of the techniques on patients are yet to be evaluated.

Based on the available literature and RCTs, it is safe to mention that untargeted microbiome-based therapeutic approaches have been more deeply explored thus far, particularly in patients with gut and liver diseases. Therefore, these interventions have more robust data to indicate their clinical effectiveness in humans than targeted approaches. However, in the future, personalized treatment plans will likely be more helpful for these patients, as they can combine microbiome-based interventions with other sophisticated omics tools, e.g., metabolomics, transcriptomics, and proteomics. A possible treatment plan in these patients may potentially include a more personalized approach based on their gut microbiome and gut-derived metabolome profiling. Targeting specific harmful microbial strains and the well-known microbiome-derived toxins (e.g., LPSs and candidalysin) or improving the production of beneficial metabolites through a precision approach could be useful and possibly increase treatment efficacy. In addition, clinical efficacies based on these treatment plans could be assessed by the established disease-specific biomarkers, histology, or microbiome-related markers, such as serum concentrations of LPSs or SCFAs, as well as fecal concentrations of *F. prausnitzii*. It is also important to note that as the microbiome signature varies widely from one individual to another, the development of microbiome-based biomarkers may require robust validation before being applied to clinical settings.

## 6. Conclusions

In conclusion, alterations in the bacterial, fungal, and viral microbiome and associated microbial products occur in all hepatobiliary and gastrointestinal diseases and may contribute to the worsening of the disease. Increased gut permeability plays a central role in developing various liver and gut diseases. Sophisticated and precise microbiome-based therapeutic approaches, including the application of bioengineered bacteria and phage therapy, show promise for gut and liver diseases. However, the applicability of preclinical findings in larger clinical settings is still poorly understood, and large RCTs are required to properly evaluate their potential in humans.

## Figures and Tables

**Figure 1 microorganisms-13-01188-f001:**
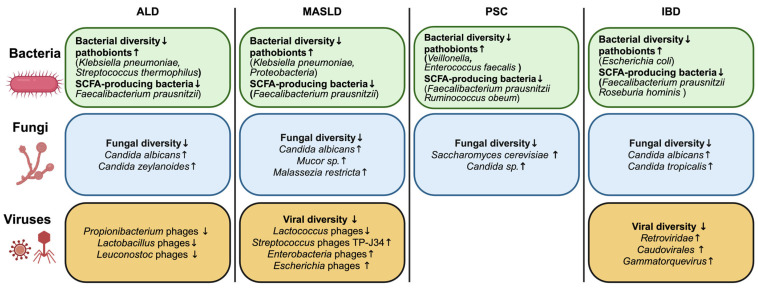
Various liver and gut diseases are associated with common aberrations of the bacterial, fungal, and viral microbiomes. A summary depicting the overall significant changes in the gut bacteriome, mycobiome, and virome compositions in chronic liver and gut diseases. ↑ denotes increase, whereas ↓ denotes decrease. ALD, alcohol-associated liver disease; IBD, inflammatory bowel disease; MASLD, metabolic dysfunction-associated steatohepatitis; PSC, primary sclerosing cholangitis. Created with a license from biorender.com.

**Figure 2 microorganisms-13-01188-f002:**
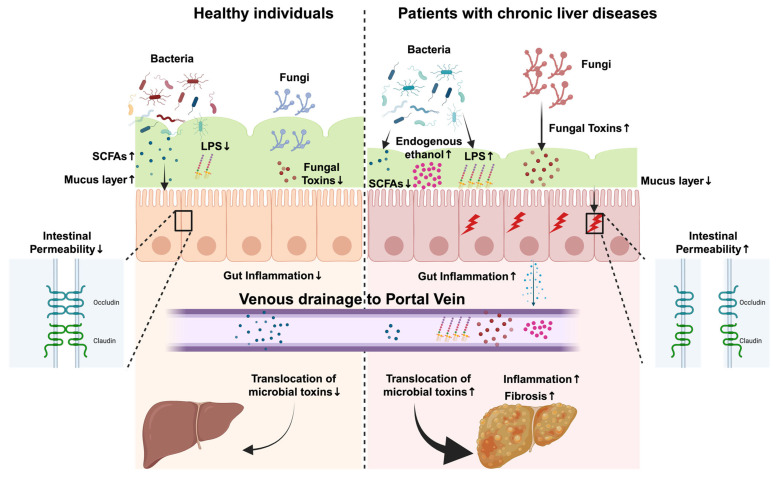
Common pathophysiological steps in chronic liver disease related to microbial products affecting the gut barrier. A comparative analysis between normal intestinal homeostasis in healthy individuals and the dysregulated state in patients with chronic liver diseases. In healthy individuals, an intact gut barrier prevents the translocation of microbial toxins into the bloodstream. Furthermore, the production of short-chain fatty acids (SCFAs) by commensal bacteria provides a protective, anti-inflammatory environment in the gut. In contrast, dysbiosis of the gut microbiome (increased abundance of pathobionts and decreased abundance of beneficial microbes, including SCFA producers) and a related increase in toxic microbial products lead to intestinal epithelial injury (e.g., disruption of tight junctions), chronic inflammation, and gut barrier dysfunction in patients with chronic liver diseases. This facilitates the translocation of microbial toxins [lipopolysaccharides (LPSs), ethanol, and fungal toxins] from the gut to the liver via the portal circulation, contributing to hepatic inflammation and fibrosis. ↑ denotes increase, whereas ↓ denotes decrease. Created with a license from biorender.com.

## Data Availability

No new data were created or analyzed in this study.

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
