# Peer review of "Impact of Gut Microbiome on Gut Permeability in Liver and Gut Diseases"

_microorganisms, 2025, doi:10.3390/microorganisms13061188_

Round 1
Reviewer 1 Report
Comments and Suggestions for Authors
comment 1) The most important thing is that authors should add a new section entitled
Exercising challenges and research directions. It is very important to mention existing challenges and explain how we can address them.
comment 2) in table 1) please state whether different Type of interventions are associated with increased concentration of butyrate or abundance of butyrate-producing bacteria involved in upregulation of TJ proteins such as claudin-1 and ZO-1
comment 3) check the text for minor grammar problems
Comments on the Quality of English Languagecheck the text for minor grammar problems
Author Response
Comments and Suggestions for Authors
comment 1) The most important thing is that authors should add a new section entitled Exercising challenges and research directions. It is very important to mention existing challenges and explain how we can address them.
-We thank the reviewer for this suggestion. We have now added a new section (Section 5) entitled “Existing challenges and future directions” to the revised manuscript.
comment 2) in table 1) please state whether different Type of interventions are associated with increased concentration of butyrate or abundance of butyrate-producing bacteria involved in upregulation of TJ proteins such as claudin-1 and ZO-1
-Thank you for this recommendation. We assessed all intervention studies, especially the randomized controlled trials. mentioned in the manuscript for reported butyrate concentrations, abundance of butyrate-producing bacteria, as well as tight junction expressions. However, only a small minority reported on butyrate-producing bacterial genera, which we included in Table 1. Butyrate and tight junction expressions were not reported in these studies.
comment 3) check the text for minor grammar problems
Comments on the Quality of English Language
check the text for minor grammar problems
-We have extensively checked the language as well as grammatical errors in the manuscript and made the necessary corrections as suggested.

Reviewer 2 Report
Comments and Suggestions for Authors
This article is a review of the effects of the gut microbiome on gut permeability, particularly in liver and intestinal disorders. This article summarizes changes in gut microbiome in different disease states and discusses in detail how these changes disrupt gut barrier function and affect disease progression. In addition, therapeutic interventions targeting the microbiome, including fecal transplants, are discussed (FMT)、 probiotics, prebiotics, synbiotics, metabiotics, and FXR agonists, among others, have demonstrated the potential to alleviate chronic disease in animal models and patients. In conclusion, this paper emphasizes the important role of intestinal microbiome in liver and intestinal diseases, and looks forward to the application of precision medicine in this field. The article is logical and well organized, but some contents need to be revised.
- Under discussion "Role of microbial metabolites and toxins in liver and gut diseases" short chain fatty acids are discussed in detail (SCFAs) and lipopolysaccharide (LPS), but other possible microbial metabolites (such as bile acids, indoles, etc.) are less discussed. It is suggested to supplement the mechanism of action of these metabolites in disease to provide a more comprehensive view.
- Conclusion of the article ("Conclusion") relatively brief, just mention "Sophisticated and precise microbiome-based therapeutic approaches... show promise for gut and liver diseases", however, the advantages and disadvantages of the various interventions mentioned in the paper are not summarized. It is recommended that the potential and challenges of these interventions and the direction of future research be summarized in more detail in the conclusion section to enhance the completeness and guidance of the article.
- Part 4 "Therapeutic interventions to alleviate liver and gut diseases" in, various interventions (e.g. Fecal Microbiota Transplantation、Probiotics, etc.) are introduced in a haphazard order, lacking a clear logical sequence (e.g., by disease type or complexity of intervention). It is suggested to reorganize these contents in logical order according to the type of disease or intervention mode, so that readers can better understand them.
- There are some repetitions in the text. For example, in the discussion "Changes in the intestinal microbiome in liver and intestinal diseases" changes in some microorganisms were mentioned several times, but no valid summaries or comparisons were made. It is recommended to merge duplicates to avoid redundancy.
- Ensure consistency of terminology throughout the document. For example, "Gut microbiome" and "Gut microbiota" although interchangeable in some cases, it is best to use them consistently in the same document to avoid confusion.
- "Fecal Microbiota Transplantation" is it miswritten as "Fecal Microbiota Transplant" (Missing "ion")
- "245.3 cases per 100,000 people" needs to be replaced by "cases per 100,000 population".
- Lines 26-28, where MASLD and MASH are juxtaposed, but MASH is actually a more serious form of MASLD. Suggested revision to “Chronic liver diseases (CLDs) encompass multiple etiologies, such as alcohol-associated liver disease (ALD) and metabolic dysfunction-associated steatotic liver disease (MASLD), which includes metabolic dysfunction-associated steatohepatitis (MASH) as a more severe form”.
- In that discussion of "microbial metabolite and toxins", it is mentioned that "SCFAs are one of the major microbial metabolites produced in the intestinal lumen by intestinal symbionts using soluble and insoluble dietary fibers" (line 257). However, the relationship between LPS and SCFAs was not clearly indicated in the discussion of LPS. It is recommended to add transitional sentences between paragraphs, such as "While SCFAs play a protective role in gut health, other microbial products such as lipopolysaccharides (LPS) can have detrimental effects".
- Part 3 Role of microbial metabolites and toxins in liver and gut diseases the tense of the whole paragraph is not consistent enough, so it is suggested to unify it into the simple present tense.
Author Response
Comments and Suggestions for Authors
This article is a review of the effects of the gut microbiome on gut permeability, particularly in liver and intestinal disorders. This article summarizes changes in gut microbiome in different disease states and discusses in detail how these changes disrupt gut barrier function and affect disease progression. In addition, therapeutic interventions targeting the microbiome, including fecal transplants, are discussed (FMT)、 probiotics, prebiotics, synbiotics, metabiotics, and FXR agonists, among others, have demonstrated the potential to alleviate chronic disease in animal models and patients. In conclusion, this paper emphasizes the important role of intestinal microbiome in liver and intestinal diseases and looks forward to the application of precision medicine in this field. The article is logical and well organized, but some contents need to be revised.
Under discussion "Role of microbial metabolites and toxins in liver and gut diseases" short chain fatty acids are discussed in detail (SCFAs) and lipopolysaccharide (LPS), but other possible microbial metabolites (such as bile acids, indoles, etc.) are less discussed. It is suggested to supplement the mechanism of action of these metabolites in disease to provide a more comprehensive view.
-We thank the reviewer for this excellent suggestion. We have now added new subsections (Section 3.5 & 3.6) where we have discussed other microbial metabolites like bile acids and indoles.
Conclusion of the article ("Conclusion") relatively brief, just mention "Sophisticated and precise microbiome-based therapeutic approaches... show promise for gut and liver diseases", however, the advantages and disadvantages of the various interventions mentioned in the paper are not summarized. It is recommended that the potential and challenges of these interventions and the direction of future research be summarized in more detail in the conclusion section to enhance the completeness and guidance of the article.
-Thank you for this recommendation. We have now added a new section (Section 5) entitled “Existing challenges and future directions” to the revised manuscript, where we have discussed the limitations and challenges of these interventions in detail.
Part 4 "Therapeutic interventions to alleviate liver and gut diseases" in, various interventions (e.g. Fecal Microbiota Transplantation、Probiotics, etc.) are introduced in a haphazard order, lacking a clear logical sequence (e.g., by disease type or complexity of intervention). It is suggested to reorganize these contents in logical order according to the type of disease or intervention mode, so that readers can better understand them.
-We thank the reviewer for this thoughtful suggestion. We have categorized Section 4 into 2 subsections (Sections 4.A. and 4.B.) based on untargeted and targeted microbiome-based therapeutic interventions to provide a more logical order. We believe that these subsections will greatly help the readers to follow the manuscript more clearly.
There are some repetitions in the text. For example, in the discussion "Changes in the intestinal microbiome in liver and intestinal diseases" changes in some microorganisms were mentioned several times, but no valid summaries or comparisons were made. It is recommended to merge duplicates to avoid redundancy.
-Thank you for this astute observation. We have checked the manuscript to remove such repetitions. Furthermore, we have added a subsection (Section 2.5 Common findings) where we have included an overall summarization of these gut microbiome-related changes in the gut and liver diseases.
Ensure consistency of terminology throughout the document. For example, "Gut microbiome" and "Gut microbiota" although interchangeable in some cases, it is best to use them consistently in the same document to avoid confusion.
-We understand the concern raised by the reviewer. Although “gut microbiome” and “gut microbiota” have slightly different meanings and were used in the manuscript accordingly in the prior version, we have now simplified the manuscript and almost exclusively used “gut microbiome” as suggested.
"Fecal Microbiota Transplantation" is it miswritten as "Fecal Microbiota Transplant" (Missing "ion")
-We thank the reviewer for detecting this mistake. We have corrected this in the revised manuscript.
"245.3 cases per 100,000 people" needs to be replaced by "cases per 100,000 population".
-Thank you for these suggestions. We have corrected this in the revised manuscript.
Lines 26-28, where MASLD and MASH are juxtaposed, but MASH is actually a more serious form of MASLD. Suggested revision to “Chronic liver diseases (CLDs) encompass multiple etiologies, such as alcohol-associated liver disease (ALD) and metabolic dysfunction-associated steatotic liver disease (MASLD), which includes metabolic dysfunction-associated steatohepatitis (MASH) as a more severe form”.
-We have rephrased this sentence in the revised manuscript as per the reviewer’s suggestion.
In that discussion of "microbial metabolite and toxins", it is mentioned that "SCFAs are one of the major microbial metabolites produced in the intestinal lumen by intestinal symbionts using soluble and insoluble dietary fibers" (line 257). However, the relationship between LPS and SCFAs was not clearly indicated in the discussion of LPS. It is recommended to add transitional sentences between paragraphs, such as "While SCFAs play a protective role in gut health, other microbial products such as lipopolysaccharides (LPS) can have detrimental effects".
-We thank the reviewer for the suggestion. We have rephrased this sentence in the revised manuscript as per the reviewer’s suggestion.
Part 3 Role of microbial metabolites and toxins in liver and gut diseases the tense of the whole paragraph is not consistent enough, so it is suggested to unify it into the simple present tense.
-We have made the necessary corrections in the revised manuscript as per the reviewer’s suggestion.

Reviewer 3 Report
Comments and Suggestions for Authors
Review of the article: "Impact of gut microbiome on gut permeability in liver and gut diseases"
The authors focus on how the gut microbiome influences gut permeability in liver and gut diseases. The article is interesting and well-written.
Here below my suggestions to further improve it.
General revision:
Typography: the authors should read their manuscript thoroughly and check: 1) space between words; 2) English of some sentences; 3) Format images with text
-Standardize the bibliography, e.g. in refs 2,3,6,8,13,14,15,18 add the other authors of the works
-Add the ref to “Figure 2” in the text
- I suggest improving the background on probiotics and prebiotics, also focusing on the mechanisms of action and the clinical role of these interventions in the management of gastrointestinal and liver diseases.
- In conclusion section, I suggest adding future perspective on how these pieces of information could help to improve patients’ life
Comments on the Quality of English Language
Minor editing is required
Author Response
Review of the article: "Impact of gut microbiome on gut permeability in liver and gut diseases"
The authors focus on how the gut microbiome influences gut permeability in liver and gut diseases. The article is interesting and well-written.
Here below my suggestions to further improve it.
General revision:
Typography: the authors should read their manuscript thoroughly and check: 1) space between words; 2) English of some sentences; 3) Format images with text
-We thank the reviewer for the suggestion. We have added the Figure 2 reference under Section 3 in the revised manuscript.
-Standardize the bibliography, e.g. in refs 2,3,6,8,13,14,15,18 add the other authors of the works
-Thank you for your recommendation. We have standardized the bibliography in the revised manuscript.
-Add the ref to “Figure 2” in the text
-Added as recommended.
- I suggest improving the background on probiotics and prebiotics, also focusing on the mechanisms of action and the clinical role of these interventions in the management of gastrointestinal and liver diseases.
-We thank the reviewer for the suggestion. We have added more background on probiotics and prebiotics and relevant mechanisms associated with these interventions in Section 4.A.2.
- In conclusion section, I suggest adding future perspective on how these pieces of information could help to improve patients’ life
-Thank you for this insightful suggestion. We have now added a new section (Section 5) entitled “Existing challenges and future directions” to the revised manuscript to discuss these more comprehensively.
Comments on the Quality of English Language
Minor editing is required
-We have extensively checked the language as well as grammatical errors in the manuscript and made the necessary corrections as suggested.

Reviewer 4 Report
Comments and Suggestions for Authors
The present article provides a comprehensive and up-to-date review of the literature on the association of the gut microbiome with intestinal permeability in the context of chronic liver and intestinal diseases. The authors provide a coherent discussion of changes in the bacterial, fungal, and viral microbiome, and present a variety of therapeutic strategies targeting the microbiota. Although the paper has many strengths, it requires several important additions and clarifications before publication.
Major Comments
1. Associations between the microbiome and disease progression are described throughout the article, but it is often unclear whether these are causal or merely correlative relationships. Please provide appropriate annotations in the text and/or in the illustrations (e.g., Figs. 1 and 2) to avoid overinterpretation.
2. The article focuses on the benefits of therapies such as FMT, probiotics, phage therapy, and FXR agonists, but completely ignores potential side effects. The therapeutics section should be supplemented with examples of such clinical risks (e.g., pathogen transmission, pruritus after FXR agonists, phage immunogenicity).
3. Intestinal permeability and inflammation are closely related to the immune response. The article should include elements such as cytokines (IL-17, IL-22, TNF-α), ILC cells, Th17 lymphocytes, etc., especially in the context of the microbiota and its metabolites.
4. It is advisable to more clearly define which of the discussed interventions show the greatest clinical potential and what data are missing to fully assess efficacy. Please also provide examples of biomarkers that could support personalization of treatment.
5. Some of the cited studies concern children and adolescents, but the authors do not distinguish pediatric results from adult data. This is worth noting or including as a limitation.
Author Response
Comments and Suggestions for Authors
The present article provides a comprehensive and up-to-date review of the literature on the association of the gut microbiome with intestinal permeability in the context of chronic liver and intestinal diseases. The authors provide a coherent discussion of changes in the bacterial, fungal, and viral microbiome, and present a variety of therapeutic strategies targeting the microbiota. Although the paper has many strengths, it requires several important additions and clarifications before publication.
Major Comments
- Associations between the microbiome and disease progression are described throughout the article, but it is often unclear whether these are causal or merely correlative relationships. Please provide appropriate annotations in the text and/or in the illustrations (e.g., Figs. 1 and 2) to avoid overinterpretation.
-Thank you for your insightful comment. We agree that changes in the gut microbiome can be more correlative in humans in particular in descriptive studies, whereas interventional studies especially in the preclinical sphere can elucidate more causal relationships between microbiome alterations and the development/exacerbation/improvement of the disease. Therefore, we have clarified these in Section 2 of the revised manuscript, which will help to avoid any overinterpretation of the findings.
- The article focuses on the benefits of therapies such as FMT, probiotics, phage therapy, and FXR agonists, but completely ignores potential side effects. The therapeutics section should be supplemented with examples of such clinical risks (e.g., pathogen transmission, pruritus after FXR agonists, phage immunogenicity).
-We thank the reviewer for this suggestion. We have now added a new section (Section 5) entitled “Existing challenges and future directions” to the revised manuscript to discuss the limitations and potential side effects.
- Intestinal permeability and inflammation are closely related to the immune response. The article should include elements such as cytokines (IL-17, IL-22, TNF-α), ILC cells, Th17 lymphocytes, etc., especially in the context of the microbiota and its metabolites.
-Thank you for this excellent suggestion. We have now elaborated more on major immune responses mediated by the microbial metabolites in the revised version of the manuscript (Section 3).
- It is advisable to more clearly define which of the discussed interventions show the greatest clinical potential and what data are missing to fully assess efficacy. Please also provide examples of biomarkers that could support personalization of treatment.
-Thank you for this astute recommendation. Therefore, we have discussed the limitations related to the interventions in the newly added Section 5 entitled “Existing challenges and future directions”. We have also mentioned the therapeutic interventions that have been evaluated more in clinical trials compared with novel interventions that have not been assessed in clinical studies yet. Furthermore, we have discussed potential microbial biomarkers that could be useful in supporting personalized treatment in patients.
- Some of the cited studies concern children and adolescents, but the authors do not distinguish pediatric results from adult data. This is worth noting or including as a limitation.
-Thank you for this suggestion. We have made the necessary corrections in the manuscript and clearly stated whether the patients in the cited studies were adults or children (Sections 2 and 4).

Round 2
Reviewer 4 Report
Comments and Suggestions for Authors
The authors have revised the manuscript accordingly to suggestions.